# Preparation of Spray-Dried Functional Food: Effect of Adding *Bacillus clausii* Bacteria as a Co-Microencapsulating Agent on the Conservation of Resveratrol

**Daniel Vázquez-Maldonado** [1], **Vicente Espinosa-Solis** [2], **César Leyva-Porras** [3], **Patricia Aguirre-Bañuelos** [4], **Fidel Martinez-Gutierrez** [1], **Manuel Román-Aguirre** [5] and **María Z. Saavedra-Leos** [5,*]

1   Facultad de Ciencias Químicas, Universidad Autónoma de San Luis Potosí, Posgrado en Ciencias Farmacobiológicas, Av. Dr. Manuel Nava No.6—Zona Universitaria, C.P., San Luis Potosí S.L.P. 78210, Mexico; lic.nut.dvm@gmail.com (D.V.-M.); fidel@uaslp.mx (F.M.-G.)

2   Coordinación Académica Región Huasteca Sur, Universidad Autónoma de San Luís Potosí, Coordinación Académica Región Huasteca Sur. km 5, Carretera Tamazunchale-San Martín, Tamazunchale S.L.P. C.P. 79960, Mexico; vicente.espinosa@uaslp.mx

3   Laboratorio Nacional de Nanotecnología (NanoTech), Centro de Investigación en Materiales Avanzados S.C. (CIMAV), Miguel de Cervantes No. 120, Complejo Industrial Chihuahua, Chihuahua C.P. 31136, Mexico; cesar.leyva@cimav.edu.mx

4   Laboratorio de Farmacología, Facultad de Ciencias Químicas, Universidad Autónoma de San Luís Potosí, Av. Dr. Manuel Nava No.6—Zona Universitaria, C.P., San Luis Potosí S.L.P. 78210, Mexico; paguirreb@uaslp.mx

5   Coordinación Académica Región Altiplano, Universidad Autónoma de San Luis Potosí, 11 Carretera Cedral Km, 5+600 Ejido San José de las Trojes Matehuala, S.L.P. C.P. 78700, Mexico; iq_manuelroman@yahoo.com

*   Correspondence: zenaida.saavedra@uaslp.mx; Tel.: +52-(488)-125-0150

**Abstract:** The effect of bacteria (*Bacillus clausii*) addition on the culturability and antioxidant activity of resveratrol prepared by spray drying was studied in this work. Inulin and lactose were employed as carrying agents and their performance compared. Resveratrol microencapsulated in inulin showed the highest antioxidant activity (26%) against free radicals. The co-encapsulated materials (bacteria and resveratrol) in inulin and lactose showed similar activities (21%, and 23%, respectively) suggesting that part of resveratrol was absorbed by the bacteria. Particles showed a regular spherical morphology with smooth surfaces, and size in the micrometer range (2–25 μm). The absence of bacteria in the SEM micrographs and the culturability activity suggested the preservation of the organisms within the micro and co-microencapsulated particles. The present work proposes the preparation of a functional food with probiotic and antioxidant properties.

**Keywords:** functional food; antioxidant activity; co-microencapsulation; spray drying; bacteria (*Bacillus clausii*) culturability

## 1. Introduction

Resveratrol (3, 4′, 5-trans-trihydroxy-stilbene) is a polyphenolic compound with antioxidant properties found in grapes, berries, peanuts, and medicinal plants. Red wine and berries juices are the most important dietary sources of resveratrol (RSV) [1]. The regular consumption of food products containing RSV may be effective in the prevention and treatment of diverse diseases such as inflammatory, cancer, aging, obesity, diabetic, cardiovascular, and neuropathies effects [2]. Additionally, it has shown antibacterial, antiviral, antifungal, and antiparasitic activity [3]. Evidently, all of these

beneficial effects have triggered the consumption of supplements and food products containing RSV. On the other hand, the use of microorganisms such as lactic acid bacteria (*Lactobacillus* spp.) and bifidobacteria (*Bifidobacterium* spp.) have shown an outstanding effect as probiotic in food products and supplements, associated with health benefits in humans by producing nutrients and cofactors, competing with pathogens for binding sites, and stimulating the immune response [4]. Health benefits include treating inflammatory bowel disease, irritable bowel syndrome, *Helicobacter pylori* infections, and antibiotic-associated diarrhea [5]. Many bioactive and probiotic ingredients are sensitive to the processing conditions and environmental factors, thus making essential the improvement of stability in order to create products with long shelf life [6]. Another important factor to consider during the study of microorganisms as ingredients is the bioactivity because as the microorganisms enter in contact with the hard environment conditions in the stomach (low pH and high concentration of bile salt in the intestine), the bioactivity is affected. However, through the microencapsulation of microorganisms, bioactivity may be preserved while the release in the gastrointestinal tract is controlled [7–9].

Microencapsulation is a technique employed in the conservation of properties of active ingredients prone to damage under certain processing or environmental conditions. Environmental conditions that may affect the activity of the ingredient include atmospheric oxygen, pH, humidity, light irradiation, and exposure to high temperature. The technique involves the use of an encapsulating material that maintains its microstructural integrity in aggressive environments where the active ingredient would lose its activity. Nutraceutical and functional ingredients such as antioxidants, vitamins, minerals, lipids, and probiotics have been microencapsulated by different methodologies [10–13]. Various substances have been used as wall materials or carrying agents for microencapsulation, including polysaccharides [14], lipids [15], and proteins [16]. Among the polysaccharides, inulin is widely employed as encapsulating material because it is a non-digestible carbohydrate polymer consisting of linear chains of fructose and glucose molecules as terminal groups linked by β-(2,1) bonds. Inulin is considered as a short-chain carbohydrate polymer because the degree of polymerization ranges between 2 and 60 repetitive units per molecule. This biopolymer is water-soluble, presented in many vegetables, fruits, and cereals [17–19]. Another carbohydrate widely used as protecting material is lactose, which consists of a galactose unit and a glucose unit chemically linked by a β-(1,4) glycosidic bond. Lactose is also water-soluble, and it constitutes 2–8% of the milk; in the solid state, it is presented as a non-hygroscopic powder with white appearance and a slightly sweet taste [20].

Several works have reported the microencapsulation of RSV for different purposes. For example, Sessa et al. [21] studied peanut oil-in-water food grade nanoemulsions of RSV encapsulated by high-pressure homogenization. Nanoemulsions of RSV with soy lecithin/sugar esters, and Tween 20/glycerol monooleate, were the most physically and chemically stable, these formulations also exhibited the highest chemical and cellular antioxidant activities, which were comparable to unencapsulated resveratrol. Koga et al. [22] microencapsulated RSV by spray drying using sodium caseinate and whey protein concentrate as carrier agents to stabilize *trans*-resveratrol. They obtained sodium caseinate microcapsules with high stability to ultraviolet (UVA) light and in vitro digestion. Peñalva et al. [23] evaluated the capability of casein nanoparticles as oral carriers for RSV. Nanoparticles prepared by spray-drying process were administered orally to rats, finding that RSV level in plasma were high and sustained for at least 8 h. The oral bioavailability of RSV when loaded in casein nanoparticles was 10 times higher than when it was administered in oral solution. Salgado et al. [24] developed a formulation for a product against the fungus *Botrytis cinerea*. First, an oil-in-water emulsion of RSV on b-glucan, lecithin or a mixture of both substances was produced. Afterwards it was dried by conventional spray drying or by Particle Gas Saturated Solution drying (PGSS). The particles formed were tested in vitro against *B. cinerea*. The b-glucan-RSV microcapsules reduced the fungal growth between 50 and 70%. Trotta et al. [25] prepared inhalable RSV by spray drying for the treatment of chronic obstructive pulmonary disease (COPD). RSV, with a spherical morphology and particle diameter less than 5 mm was successfully obtained. The cytotoxicity results of RSV on Calu-3 revealed that the cells may tolerate a high concentration of resveratrol (up to 160 mM). The expression of

interleukin-8 (IL-8) from Calu-3 cells induced with tumor necrosis factor alpha (TNF-a), transforming growth factor beta (TGF-b1), and lipopolysaccharide (LPS) were significantly reduced after treatment with spray-dried RSV. The antioxidant assay showed spray-dried resveratrol to possess an equivalent antioxidant property as compared to vitamin C.

Considering the importance of probiotics and bioactive compounds in human health, and in the food industry, the objective of the present work is to obtain a functional food by spray drying. The functional food comprises the co-encapsulation of *Bacillus clausii* and resveratrol employing protecting agents such as inulin and lactose. These agents were selected based on their functionality, since besides the encapsulating function they may act as substrates for the metabolism of the probiotic. This work also contributes to understanding the synergistic effect of probiotics on the conservation of antioxidants.

## 2. Materials and Methods

### 2.1. Materials

Inulin (99.9%), and α-Lactose monohydrate, (99%, αL·$H_2O$) were purchased from Sigma-Aldrich Chemical Co. (Toluca, Edo de México, México). Bacillus bacteria strain (*Bacillus clausii*) in Sunuberase solution was purchased from Sanofi-Aventis de México, S.A. de C.V. (Coyoacán, CDMX, México), while resveratrol (99%) was acquired from Química Farmacéutica Esteroidal S.A. de C.V., (Tláhuac, CDMX, México).

### 2.2. Preparation of Spray-Dried Functional Foods

Spray drying was employed in the preparation of the microencapsulated and co-microencapsulated powders of *Bacillus clausii* and resveratrol. Spray drying conditions were similar to those reported by Saavedra-Leos et al. [26]. Typically, the preparation of feeding solutions consisted of mixing 20 g of the corresponding carrying agent (inulin or lactose), 10 g of resveratrol, 5 mL of the commercial solution with bacteria (equivalent to a concentration of $2 \times 10^{12}$ CFU), and distilled water for obtaining a total volume of 100 mL of solution. Microencapsulation was carried out in a Mini Spray Dryer B290 (BÜCHI, Labortechnik AG, Flawil, Switzerland) at the following operation conditions: feed temperature of 40 °C, feeding flow of 7 cm$^3$/min, hot airflow of 28 m$^3$/h, aspiration of 70%, and pressure of 1.5 bar. The inlet and outlet temperatures were set as 210 and 70 °C, respectively. The parameters used in this process have been reported previously [14,27,28]. The obtained powders were weighted and labeled according to their content, where *Bc* stands for Bacillus bacteria, RSV for resveratrol, IN for inulin, and L for lactose. Powders were individually placed in airtight containers, stored in darkness at 4 °C.

### 2.3. Culturability of Bacillus clausii in the Microencapsulated Functional Food

The number of available bacteria cells was evaluated in the microencapsulated samples by means of the plate extension technique, with Trypticase-Soy Agar (TSA) (Beckton Dickinson, Germany), which has been previously used in the growth of *B. clausii* [29], using serial dilutions of the encapsulated samples from $1 \times 10^{-1}$ to $1 \times 10^{-7}$. Growing conditions were aerobic, with an incubation period of 48 h at 37 °C in a Novatech incubator (Jalisco, Mexico). Bacillus bacteria strain (*Bacillus clausii*) in Sunuberase solution was used as a control. For the determination of the number of colony-forming unit per gram (CFU/g), the concentrations exhibiting between 300 and 30 CFU ($1 \times 10^{-4}$ and $1 \times 10^{-5}$) were selected. Equation (1) was employed for the quantification of the culturability. All experiments were conducted in triplicate, and the reported values represent the average of the calculated values.

$$\frac{\text{CFU}}{\text{g}} = \left[ \frac{\text{N}° \text{ plate colonies} * \text{dilution factor}}{\text{mL sample seeded}} \right] \tag{1}$$

### 2.4. Radical Scavenging Activity of the Functional Food

The antioxidant capacity of resveratrol was determined according to the methodology reported by Hao et al. [30]. The microencapsulated powder was mixed in a solution containing 2,2-diphenyl-1-picrilhydrazyl (DPPH*) as free-radical. Typically, 1.7 mL of DPPH* ethanol solution (0.1 mmol/L) was added to 1.7 mL of microencapsulated sample diluted in ethanol. Initial powder concentrations solutions tested were 2.5, 5, and 15 µg/mL. The mixed solution (3.4 mL) was poured into a 10 mm thick quartz cell. The antioxidant capacity was evaluated as the decrease in the initial concentration of DPPH* scavenged by resveratrol after 30 min of preparation. The variation in the absorbance intensity of DPPH* was measured at a wavelength of 537 nm in a UV-Vis spectrophotometer Evolution 220 (Mettler Toledo, Powai Mumbai, India). The scavenging activity (%DPPH*) was calculated according to Equation (2):

$$\text{Scavenging activity}\left(\%\text{DPPH}^*\right) = \frac{A_0 - A_{30}}{A_0} \times 100 \tag{2}$$

where $A_0$ is the initial absorbance of DPPH*, and $A_{30}$ is the absorbance of the DPPH* after 30 min of adding the microencapsulated antioxidant.

### 2.5. Scanning Electron Microscopy (SEM) of the Microencapsulated Powders

A scanning electron microscope SU3500 (Hitachi, Japan) operated at 15 kV in low vacuum conditions of 60 Pa, and with a backscattered detector (BSE), was employed to observe the morphology of the particles. Each sample was dispersed on double-side carbon film and observed without any further treatment.

### 2.6. Statistical Analysis

All experiments were performed in triplicate, reporting mean values and standard deviations. One-way analysis of variance (ANOVA) was performed to establish a significance level of 0.05, and the Tukey's honestly significant difference (HSD) post hoc test was used to determine the difference between means. The statistical analyses were conducted using the IBM SPSS statistics version 21.0 software (SPSS Inc., Chicago, IL, USA).

## 3. Results and Discussion

### 3.1. Culturability of Spray Dried Functional Foods

Spray drying is widely employed as a technique for the encapsulation of active ingredients and drying heat-sensitive compounds, in both, pharmaceutical and industrial contexts [31–34]. Although, it is typically considered as a dehydration process, has also been employed as an encapsulation technique. The drying efficiency is an important factor to consider, since the higher the powder recovery, the better the drying efficiency [35]. In this case, a similar amount of powder was recovered in the cyclone for all microencapsulated and co-microencapsulated samples, which was about 8 g per run. The initial amount of dry matter was 30 g (20 g of carrier agent and 10 g of resveratrol), thus the calculated average yield was 26%. The mass losses may be attributed to effect of the different parameters of spray drying such as inlet temperature and air flow, as well as to the low Tg of carrying agents [36,37].

Figure 1 shows the effect of inulin and lactose on the culturability in the microencapsulation of *B. clausii* cells in samples *Bc*IN and *Bc*L, respectively; the number of viable cells is expressed in Colony-Forming Units per gram (CFU/g). The results indicated that after the spray drying process all samples showed a significant decrease in the culturability compared to the control $10.28 \pm 0.03$ Log10 CFU/g ($p < 0.05$). There were not significant statistical differences between samples of *B. clausii* microencapsulated in inulin (*Bc*IN = $8.68 \pm 0.01$ Log10 CFU/g) and lactose (*Bc*L = $8.66 \pm 0.07$

Log10 CFU/g) ($p > 0.05$). This suggested that both wall materials exerted a similar thermo-protective effect on *B. clausii* cells during the spray drying process. This could be attributed to the ability of hydroxyl groups, presented in lactose and inulin, to form hydrogen bonding interactions with the membrane proteins of *B. clausii*, that during the dehydration process, prevented the denaturation of proteins and retained the native integrity [38]. Furthermore, *B. clausii*, in its sporulated form, has greater resistance to high temperatures [39]. There are several studies of encapsulation of probiotic agents by spray drying, employing different matrices for protecting the microorganisms from thermal degradation. For example, Araujo-Uribe et al. [40] found greater viability in *Bacillus polymuxa* spores in comparison with *Lactobacillus delbruekii* when subjected to spray drying encapsulation, using 35% of maltodextrin as carrier agent, and an aspiration percentage of 60%. Utami et al. [41], demonstrated that the use of maltodextrin improved the viability of Bacillus NP5 when subjected to spray drying at inlet and outlet temperatures of 120 and 70 °C, respectively.

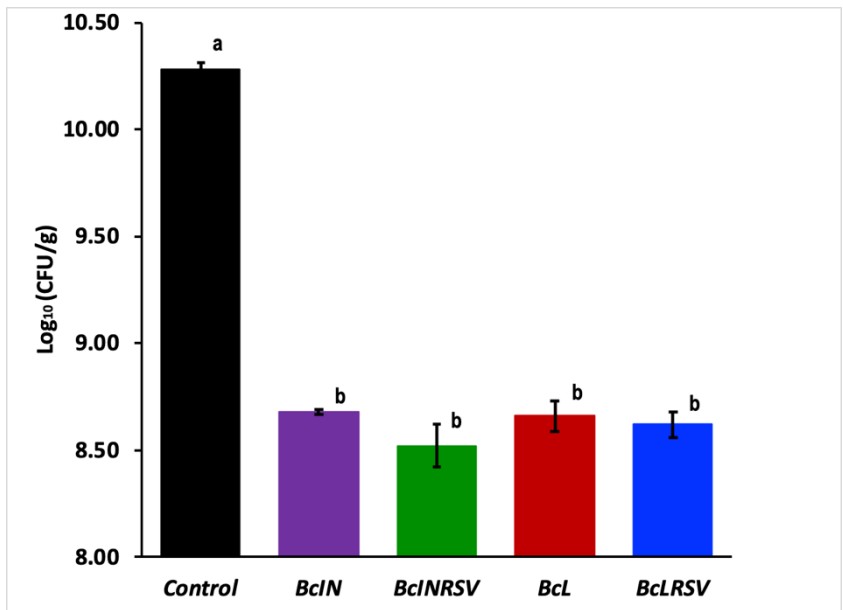

**Figure 1.** Culturability of *B. clausii* (control) spray-dried in inulin and lactose matrices without resveratrol (*Bc*IN and *Bc*L) and with resveratrol (*Bc*INRSV, and *Bc*LRSV), a and b are the parameters of the Tukey's honestly significant difference (HSD) test.

Romano et al. [42] evaluated the effect of crystalline and amorphous inulin as protective matrices of *Lactobacillus plantarum* CIDCA 83114, during the spray drying process at inlet and outlet temperatures of 160 and 65 °C, respectively. They found that crystalline inulin showed greater stability at high values of water activity ($a_w$) and as the storage temperature approached the Tg. Raddatz et al. [43] evaluated the encapsulation efficiency of *lactobacillus acidophilus* in emulsions made of pectin and prebiotics such as inulin, Hi-Maize starch, and rice bran. They reported a plasticizing effect of inulin that favors the development of stable networks with pectin, preserving the encapsulation of the microorganism after 90 days.

Yoha et al. [7] found that *Lactobacillus plantarum* microencapsulated with fructooligosaccharides (FOS) by spray drying, improved the encapsulation efficiency and preserved 96% of viability. Maleki et al. [44] established that in a system with 57.2% of whey protein, 25% crystalline nanocellulose, and inulin concentration of 17.78%, the encapsulation efficiency and viability of *Lactobacillus rhamnosus* increased. This was attributed to the conservation of the osmotic pressure in the intracellular environment of bacteria, and to the interaction of proteins with polysaccharides. Similar results have been reported by other authors, that have evaluated the effect of carrying agents such as starch, cellulose, pectin, carrageenan, inulin, and lactose on the viability of probiotics such as *Bifidobacterium infantis*

and *Lactobacillus acidophilus* [45–48]. Some investigations have used lactose, to encapsulate functional ingredients and probiotic microorganisms. Li et al. [49] found that lactose with whey protein in a ratio of 4:1 showed the highest encapsulation efficiency of ethyl butyrate. Similar results were reported by other authors, however, the use of lactose as carrying agent implies some limitations because there is a significant group of the population showing intolerance to this polysaccharide [50–52]. Tantratian and Pradeamchai [38] evaluated five carbohydrates used as protective matrices for *Lactobacillus plantarum FT 35*, during the encapsulation process by spray drying. These authors reported that wall materials with Tg higher than 100 °C provided a greater number of viable cells. Lactose with Tg of 119.3 °C, presented greater protection than glucose and sucrose, while no significant statistically differences were found for cells encapsulated in maltodextrin and soluble starch matrices. Additionally to the characteristic of protective agent during drying, lactose showed the ability of stabilize the cell membrane proteins of the probiotic agent.

Additionally, Figure 1 shows the effect of inulin and lactose on the culturability of co-microencapsulates of *B. clausii* with resveratrol. There were no significant statistical differences in the co-microencapsulation of *Bc* and RSV using both carrier agents: *Bc*INRSV and *Bc*LRSV samples showed $8.52 \pm 0.10$ and $8.62 \pm 0.06$ Log10 CFU/g, respectively. However, compared to the microencapsulated counterparts (*Bc*IN and *Bc*L), there is a decrement in the number of viable cells, which is related to the inhibitory effect of resveratrol on some microorganisms. Mora-Pale et al. [53] reported the antimicrobial effect that resveratrol can exert on various bacteria, including Bacillus cereus. Ma et al. [54] mentioned that resveratrol has an antibacterial effect against pathogens, caused by the ability of inhibiting the electron transport, observed as the decrease in the proliferation of microorganisms, as well as inhibiting cell division. The results reported herein indicated that the concentration of *B. clausii* in the microencapsulated and co-microencapsulated powders (*Bc*IN, *Bc*L, and *Bc*INRSV, *Bc*LRSV) may be considered as functional foods because are in the concentration level required to exert a probiotic effect, ($>1 \times 106$ CFU/g) [55].

## 3.2. Radical Scavenging Activity

Figure 2 shows the effect of inulin and lactose on the antioxidant activity of microencapsulated samples of *B. clausii* and resveratrol. The samples *Bc*IN and *Bc*L presented the lowest antioxidant activity values compared to the samples containing resveratrol. Pasqualetti et al. [56] reported that single spray dried inulin showed antioxidant capacity of 0.8 nmolTE/mg. It is worth mentioning that the antioxidant activity of inulin reported by these authors was determined in samples in contact with colon cells. Thus, the antioxidant activity of inulin could be attributed to the reactivity of colon cells with reactive oxygen species (ROS), rather than directly to the polysaccharide. Shang et al. [57] showed that antioxidant activity of single inulin is minimal, while when administered in vivo, the model significantly increased. Evidently, the antioxidant activity values reported herein are relatively higher. Furthermore, antioxidant activity of single lactose has been attributed to the ability of forming Maillard products with proteins [58]. Lactose as reducing carbohydrate may interact with proteins of the spore membrane of *B. clausii* and form a minimum amount of Maillard products during the spray drying process. These observations agreed with those obtained in this study, where the microencapsulated *Bc*IN presented higher antioxidant activity than *Bc*L.

The effect of inulin and lactose on the antioxidant activity of resveratrol microencapsulates (samples INRSV and LRSV, respectively) is also shown in Figure 2. Sample INRSV showed a higher antioxidant activity than LRSV. This may be attributed to the protecting role of inulin as encapsulating material of bioactive compounds, as demonstrated by Silva et al. [59]. Ha et al. [60] evaluated the effect of whey protein and inulin on the physicochemical and prebiotic properties of resveratrol nano-encapsulates prepared by ionic gelation methods, and found that an increase in the concentration of inulin from 1 to 3% improved the encapsulation efficiency from 79 to 83%. Although to a lesser extent, lactose can also stabilize and protect active compounds by encapsulation, conserving some antioxidant activity [9]. In the microencapsulation processes of resveratrol, inulin was a better wall

material than lactose, avoiding the interconversion of *trans*-resveratrol into the inactive counterpart *cis*-resveratrol.

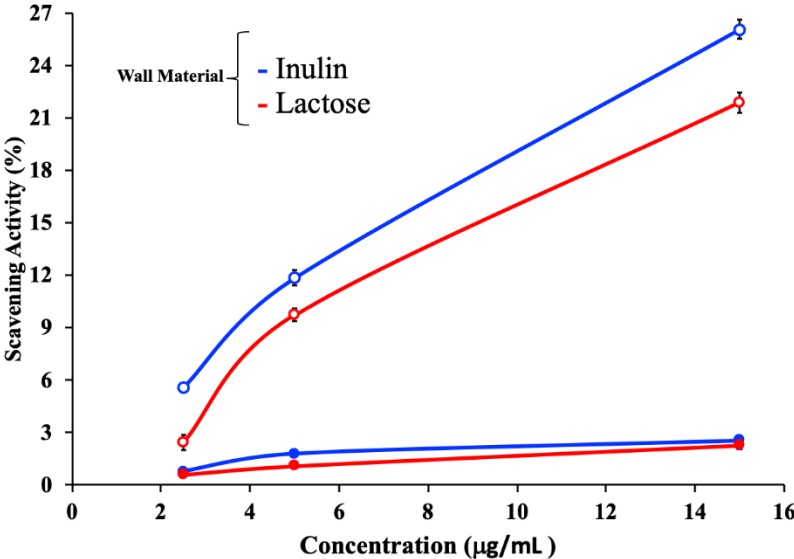

**Figure 2.** Effect of inulin and lactose matrices on the scavenging activity of microencapsulated *B. clausii* (BcL and BcIN) closed circles; and resveratrol (LRSV and INRSV) open circles.

Regarding the antioxidant activity of the co-microencapsulated *B.clausii* with resveratrol samples (Figure 3), lactose (*Bc*LRSV) showed a slight higher antioxidant activity compared to inulin co-microencapsulates (*Bc*INRSV). Clearly, this behavior may be attributed to a synergistic effect established between the components of the co-microencapsulates [61]. However, *B. clausii* may act as a fermenting agent, producing phenolic compounds [29], hydrolyzing proteins and generating active antioxidant peptides [62]. For example, bacteria of the genus *Bacillus* may generate antioxidant compounds such as carotenoids or riboflavin [63]. Therefore, the aforementioned synergism may be caused by the development of a defense mechanism of the bacteria against external agents such as free radicals (DPPH*), resveratrol and the carrying agent [64,65].

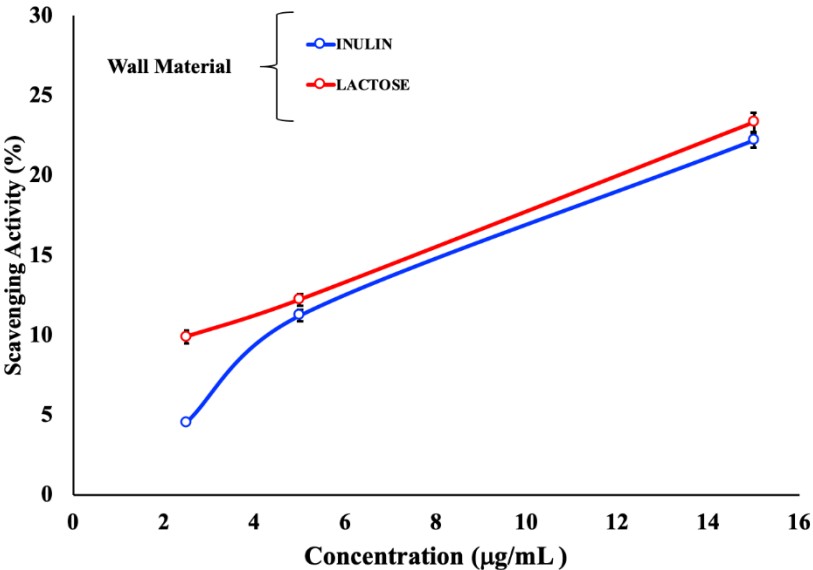

**Figure 3.** Effect of inulin and lactose matrices on the scavenging activity of co-microencapsulated *B. clausii* with resveratrol (*Bc*LRSV and *Bc*INRSV).

Figure 4 compares the scavenging activity at the maximum powder concentration tested in this study (15 µg/mL). The higher antioxidant activity was observed for sample INRSV. With exception of *Bc*In and *Bc*L samples that showed minimal activities, the rest of the samples (LRSV, *Bc*INRSV, and *Bc*LRSV) presented no significant statistical differences on the scavenging activity. The observed decrease in the antioxidant activity of *Bc*INRSV co-encapsulates could be attributed to the adsorption capacity of Bacillus spores, which are reported that retain soluble compounds inside the walls [66,67]. Spirizzi et al. [68] reported a correlation between the antioxidant activity of hydrogels polymers and the degree of crosslinking. These soluble compounds may modify the polymeric networks of microencapsulates, keeping resveratrol inside the particle and reducing its availability for interacting with the surroundings of powder particle.

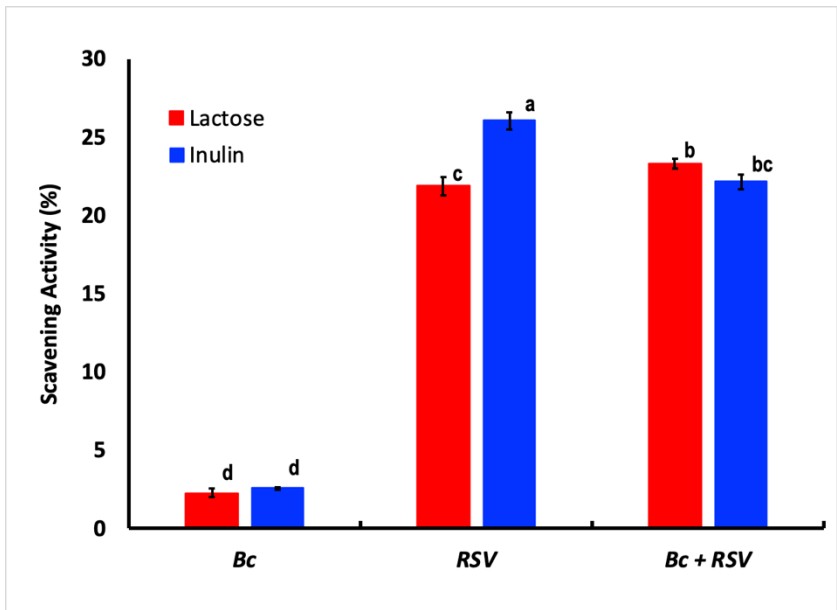

**Figure 4.** Effect of the microencapsulation of *B. clausii* (*Bc*), resveratrol (RSV) and Co-microencapsulation of (*Bc* + RSV) processes on the scavenging activity of spray dried powders at concentration of 15 mg/mL. a, b, bc, c and d are the parameters of the Tukey's HSD test.

### 3.3. Scanning Electron Microscopy (SEM)

Scanning electron microscopy (SEM) is a technique that allows evaluating the microstructural characteristic and particle morphology of powders. Figure 5 shows the SEM micrographs of microencapsulated and co-microencapsulated samples. *Bc*IN and *Bc*L microencapsulated samples formed spherical particles with dimensions in the range of 2–25 µm. In addition to the spherical shape particles, microencapsulated samples with resveratrol (INRSV and LRSV) developed some rod-like microstructures with length of 40–60 µm and about 10 µm in diameter. These particles were identified as microencapsulates rich in resveratrol [60]. The co-microencapsulation of *B. clausii* and resveratrol (*Bc*INRSV and *Bc*LRSV), developed some irregular shape microstructures. According to Berta Nogeiro et al. [33] the relatively high viscosity of chitosan promotes the formation of irregular shape particles in the microencapsulated. Thus, because inulin fibers increase the viscosity of the feeding solution, the subsequent spray drying process may be disturbed, promoting the formation of irregular shape particles. Littringer et al. [69] evaluated the effect of the spray drying outlet temperature of mannitol systems on the surface morphology of particles. They found rough surfaces at outlet temperature of 67 °C and smooth surfaces at temperature of 102 °C. Tobin et al. investigated the effect of inulin on the microstructural properties of whey protein and lactose particles, and reported that samples with lactose presented brittle spherical structures, and as inulin content increased the presence of brittle morphologies tend to disappear [70]. In general, all the samples prepared herein showed a smooth

surface while the recorded outlet temperature was 70 °C. Unfortunately, SEM micrographs did not show clear evidence of the state of *B. clausii* bacteria inside or outside the micro and co-encapsulates. However, the culturability results demonstrated the presence of live bacteria encapsulated after the drying process. This observation indicates that some bacteria are indeed being encapsulated and preserved within the carrier agent, while other bacteria that remained outside the encapsulating material was thermally degraded during spray drying, and for this reason were not observed in micrographs. Nevertheless, additional studies are necessary to demonstrate this argument.

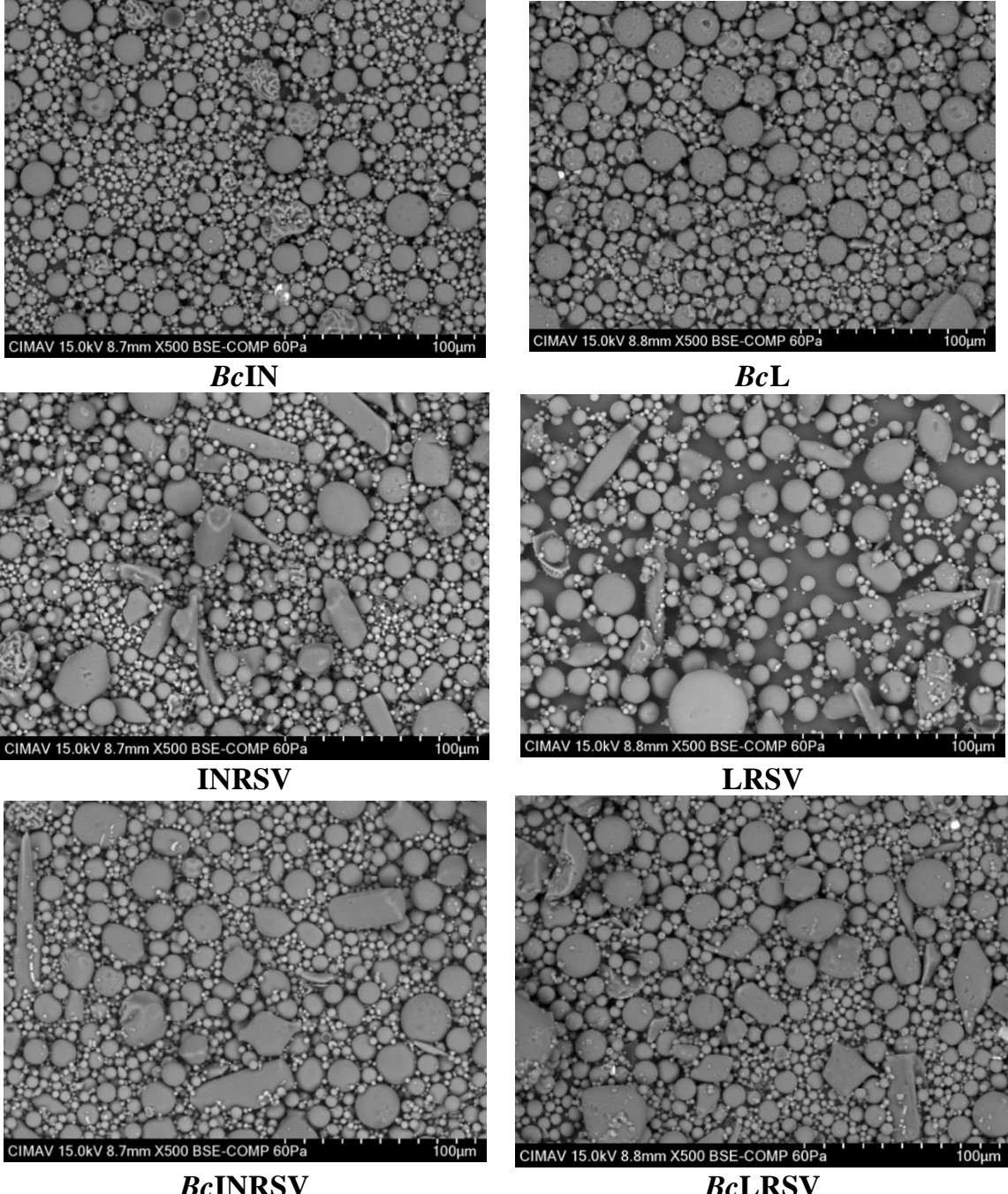

**Figure 5.** SEM micrographs of powders: microencapsulated *B. clausii* (*Bc*IN, *Bc*L), resveratrol (INRSV, LRSV), and co-microencapsulated (*Bc*INRSV, *Bc*LRSV).

## 4. Conclusions

A functional food with antioxidant and probiotic properties was prepared by spray drying. Inulin and lactose were employed as carrying agents in the micro and co-microencapsulation of the resveratrol and *Bacillus clausii* bacteria. Dried powders showed bacterial activity (culturability) indicating that organisms were successfully encapsulated within the carrying agents. Resveratrol microencapsulated in inulin showed the highest antioxidant activity, while the co-microencapsulates containing bacteria and resveratrol showed similar activity. Apparently, the bacteria absorbed resveratrol reducing its availability in the vicinity of the particle. Particle size and morphology showed particles with smooth surface in the micrometer size range. The absence of bacteria in SEM micrographs and the culturability results indicated the preservation of living organisms inside the carrying agent after the spray drying process. From the two carrying agents tested, Inulin showed a better performance in the microencapsulation of resveratrol, while for the co-microencapsulation of resveratrol and *B. clausii*, both wall materials showed similar results.

**Author Contributions:** Conceptualization, M.Z.S.-L. methodology, M.R.-A.; validation, C.L.-P. and M.Z.S.-L.; formal analysis, D.V.-M.; investigation, D.V.-M.; writing—original draft preparation, V.E.-S. and C.L.-P.; writing—review and editing, V.E.-S., C.L.-P. and M.Z.S.-L.; visualization, V.E.-S.; supervision, P.A.-B. and F.M.-G. All authors have read and agreed to the published version of the manuscript.

**Funding:** This research received no external funding.

**Acknowledgments:** Daniel Vázquez-Maldonado is grateful to the Consejo Nacional de Ciencia y Tecnologia (CONACYT) in Mexico for the financial support provided during his Ph.D. Studies through the scholarship No. 775205. And the technical assistance of Juan Manuel Delgado Cervantes, from Facultad de Medicina of the Universidad Autónoma de San Luis Potosí.

**Conflicts of Interest:** The authors declare no conflict of interest.

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
