# Peer review of "Preparation of Spray-Dried Functional Food: Effect of Adding Bacillus clausii Bacteria as a Co-Microencapsulating Agent on the Conservation of Resveratrol"

_processes, doi:10.3390/pr8070849_

Round 1

Reviewer 1 Report

Title:

 Preparation of spray-dried functional food: effect of adding Bacillus clausii bacteria as a co-microencapsulating agent on the conservation of resveratrol

GENERAL COMENTS:

The manuscript presents very interesting research about possibilities of microencapsulation of resveratrol adding Bacillus clausii bacteria via spray drying. Adding Bacillus clausi bacteria for preparation resveratrol is innovative in this study. Some other research were done to prepare resveratrol by spray drying but not mentioned in this research (e.g. Koga et al. 2016; Penalva et al. 2018; Salgado et al. 2015; Trotta et al. 2015), therefore it is necessary to improve introduction section and explain what new method is proposed in this study. Some other suggestions are showed below.

INTRODUCTION SECTION

Introduction section is rather well organized but please add some other research on spray drying of resveratrol explain in detail what new aspect is done in this study.

When you cite References as Author et al., please add reference number after et al., [23], not at the end of sentence.

MATERIALS AND METHODS

Line 97-100:

Regarding sentence: “Typically, the preparation of feeding solutions consisted of mixing 20 g of the corresponding carrying agent (inulin or lactose), 10 g of resveratrol, 5 mL of the commercial solution with bacteria (equivalent to a concentration of 2x1012  CFU), and distilled water for obtaining a total volume of 100 mL of solution” please add some explanation about procedure or recipe used in this study – it is your own methodology or it is based on earlier study. If it is based on earlier study, please cite some references.

Line 100-103:

Regarding sentence: “Microencapsulation was carried out in a Mini Spray Dryer B290 (Buchi, Switzerland) at the following operation conditions: of  feed temperature of 40 ºC, feeding flow of 7 cm3/min, hot airflow of 28 m3/h, aspiration of 70%, and  pressure of 1.5 bar. The inlet and outlet temperatures were set as 210 and 70 ºC, respectively...” Please mention some explanation about parameters used in this study or cite some

 references.

Line 104-106

Please explain in detail symbols used in this study, what do you mean control – it is resveratrol before spray drying? This symbols should be also explained in figure legends.

Line 109-110:

Please cite references for the number of available bacteria cells was evaluated by means of the plate extension technique, with Trypticase-Soy Agar (TSA) if available.

Statistical analysis

Authors performed only ANOVA, one way analysis of variance I suggest to add some post host test result showing different letters for statistically different means in figure 1 and 3.

RESULTS

Results and discussion are rather well described. Please mention some results about drying yield.

Please improve statistical differences presentation in figures 1 and 3.and add symbol explanation after figures.

Proposed additional references:

Koga, C. C., Andrade, J. E., Ferruzzi, M. G., & Lee, Y. (2016). Stability of Trans-Resveratrol Encapsulated in a Protein Matrix Produced Using Spray Drying to UV Light Stress and Simulated Gastro-Intestinal Digestion. Journal of Food Science, 81(2), C292–C300. https://doi.org/10.1111/1750-3841.13176

Peñalva, R., Morales, J., González-Navarro, C. J., Larrañeta, E., Quincoces, G., Peñuelas, I., & Irache, J. M. (2018). Increased oral bioavailability of resveratrol by its encapsulation in casein nanoparticles. International Journal of Molecular Sciences, 19(9). https://doi.org/10.3390/ijms19092816

Salgado, M., Rodríguez-Rojo, S., Alves-Santos, F. M., & Cocero, M. J. (2015). Encapsulation of resveratrol on lecithin and β-glucans to enhance its action against Botrytis cinerea. Journal of Food Engineering, 165, 13–21. https://doi.org/10.1016/j.jfoodeng.2015.05.002

Trotta, V., Lee, W. H., Loo, C. Y., Haghi, M., Young, P. M., Scalia, S., & Traini, D. (2015). In vitro biological activity of resveratrol using a novel inhalable resveratrol spray-dried formulation. International Journal of Pharmaceutics, 491(1–2), 190–197. https://doi.org/10.1016/j.ijpharm.2015.06.033

Reviewer 2 Report

The authors described important issues in the examined area of preparing a functional food with probiotic and antioxidant properties. The paper contains new contribution compared to already published works.

The strong point of the paper: it offers an effective approach to deal with spray-dried functional food – the authors analyze effect of adding Bacillus clausii bacteria as a co-microencapsulating agent on the conservation of resveratrol. These new findings significantly contribute to better understanding of the investigated problem domain.

The approach is grounded on solid knowledge, and the results are important for practice. The proposed approach is successful in preparing functional food with antioxidant and probiotic properties by spray drying - dried powders showed bacterial activity, and it  indicates that organisms were successfully encapsulated within the carrying agents. Resveratrol microencapsulated in inulin showed the highest antioxidant activity, and the co-microencapsulates (which contained bacteria and resveratrol) showed similar activity. As the bacteria absorbed resveratrol, its availability on the vicinity of the particle was reduced. The authors used an advanced technique of micrographs to evaluate the microstructural characteristic and particle morphology of powders, based on  scanning electron microscopy. It enabled them to conclude that living organisms inside the carrying agent after the spray drying process were preserved.

Introduction provides relevant and understandable information about the problem addressed and about motivation for such study, but instead of a clear statement of the objective of the paper there is only a sentence: “this work presents the preparation of a functional food by spray drying. The article provides a comprehensive survey of the recent developments relevant to the topic of the paper, and the list of references is vast and impressive. The authors stressed an importance of developing appropriate procedures for preparing functional food with probiotic and antioxidant properties.

Methods were selected and discussed adequately, results and discussion are presented correctly, conclusions are concise and relate back to objectives. However, it would be clearer to follow the authors’ reasoning if additional figures were added.

Several issues should be addressed by the authors before it is printed:

  1. Clear statement of the objective of the paper in Introduction.
  2. Enhancement of discussion of results with an additional figure.
  3. Enhancement (support) of Conclusions with some specifics mentioned in previous section.

Author Response

REVIEWER 2

The authors described important issues in the examined area of preparing a functional food with probiotic and antioxidant properties. The paper contains new contribution compared to already published works.

The strong point of the paper: it offers an effective approach to deal with spray-dried functional food – the authors analyze effect of adding Bacillus clausii bacteria as a co-microencapsulating agent on the conservation of resveratrol. These new findings significantly contribute to better understanding of the investigated problem domain.

The approach is grounded on solid knowledge, and the results are important for practice. The proposed approach is successful in preparing functional food with antioxidant and probiotic properties by spray drying - dried powders showed bacterial activity, and it  indicates that organisms were successfully encapsulated within the carrying agents. Resveratrol microencapsulated in inulin showed the highest antioxidant activity, and the co-microencapsulates (which contained bacteria and resveratrol) showed similar activity. As the bacteria absorbed resveratrol, its availability on the vicinity of the particle was reduced. The authors used an advanced technique of micrographs to evaluate the microstructural characteristic and particle morphology of powders, based on  scanning electron microscopy. It enabled them to conclude that living organisms inside the carrying agent after the spray drying process were preserved.

Introduction provides relevant and understandable information about the problem addressed and about motivation for such study, but instead of a clear statement of the objective of the paper there is only a sentence: “this work presents the preparation of a functional food by spray drying. The article provides a comprehensive survey of the recent developments relevant to the topic of the paper, and the list of references is vast and impressive. The authors stressed an importance of developing appropriate procedures for preparing functional food with probiotic and antioxidant properties.

Methods were selected and discussed adequately, results and discussion are presented correctly, conclusions are concise and relate back to objectives. However, it would be clearer to follow the authors’ reasoning if additional figures were added.

Several issues should be addressed by the authors before it is printed:

  1. Clear statement of the objective of the paper in Introduction.

Answer: Taking into consideration the reviewer´s advice, we made clear our objective in the introduction section. See lines: 98.

  1. Enhancement of discussion of results with an additional figure.

Answer: In order to add another figure, we had to split our original figure 2 into two new figures. See changes as follows:

  • Figure 2: lines 252-253

  • Figure 3: Lines 276-277

  • Discussion: Lines 263-265

  1. Enhancement (support) of Conclusions with some specifics mentioned in previous section.

Answer: conclusions were modified. Lines 319-329